

# Transcriptional factor regulation network and competitive endogenous RNA (ceRNA) network determining response of esophageal squamous cell carcinomas to neoadjuvant chemoradiotherapy

Mingrui Shao and Wenya Li

Department of Thoracic Surgery, The First Affiliated Hospital of China Medical University, Shenyang, Liaoning, China

## ABSTRACT

**Background**. Neoadjuvant chemoradiotherapy (nCRT) followed by surgery benefits survival for patients with esophageal squamous cell carcinomas (ESCC) compared with surgery alone, but the clinical outcomes of nCRT are heterogeneous. This study aimed to elucidate transcriptional factor (TF) regulation network and competitive endogenous RNA (ceRNA) network determining response of ESCC to nCRT.

**Materials and Methods**. RNA microarray data of GSE59974 and GSE45670 were analyzed to investigate the significant changes of lincRNAs, miRNAs, mRNAs in responders and non-responders of nCRT in ESCC. Functional and enrichment analyses were conducted by clusterProfiler. The target lincRNAs and mRNAs of miRNAs were predicted by miRWalk. The ceRNA and TF regulatory networks were constructed using Cytoscape.

**Results**. Differentially expressed genes between responders and non-responders mainly enriched in biological process including Wnt signaling pathway and regulation of cell development and morphogenesis involved in differentiation. Besides, these genes showed enrichment in molecular function of glycosaminoglycan binding, metalloendopeptidase inhibitor and growth factor activity. KEGG analysis enriched these genes in pathways of neurotrophin signaling pathway, cell adhesion molecules and Wnt signaling pathway. We also constructed ceRNA network and TF network regulating response of ESCC to nCRT. Core regulatory miRNAs were miR-520a, miR-548am, miR-3184, miR-548d, miR-4725, miR-148a, miR-4659a and key regulatory TFs included MBNL1, SLC26A3, BMP4, ZIC1 and ANKRD7.

**Conclusion**. We identified significantly altered lincRNAs, miRNAs and mRNAs involved in the nCRT response of ESCC. In addition, the ceRNA regulatory network of lincRNA-miRNA-mRNA and TF regulatory network were constructed, which would elucidate novel molecular mechanisms determining nCRT response of ESCC, thus providing promising clues for clinical therapy.

Corresponding author
Wenya Li, wenyali@cmu.edu.cn,
saint5288@hotmail.com

## INTRODUCTION

As a common malignant tumour of upper digestive tract, esophageal squamous cell carcinoma (ESCC) represents a significant health burden worldwide due to its aggressive growth (*Kang et al., 2015*). There is accumulating evidence that surgical resection of locally advanced ESCC is an effective method to control the progression of this disease (*Sjoquist et al., 2011*). In some cases, however, the recurrence after curative resection and unsatisfactory survival status still poses a significant obstacle for surgeons (*Rohatgi et al., 2006*). In recent years, it has been proved that neoadjuvant chemoradiotherapy (nCRT) followed by surgery benefits survival for patients with ESCC compared with surgery alone, which is recommended in the guidelines for ESCC therapy (*Allum et al., 2011*). Currently, up to one-third of patients show a pathologically complete response after neoadjuvant chemoradiotherapy. To optimise the efficacy of neoadjuvant treatment for individual patients, prediction of response to neoadjuvant treatment is highly desired (*Eyck et al., 2018*).

Noncoding RNAs (ncRNAs) are transcripts which possess no protein coding potential, the number of which constitutes over 98% of the entire genome transcripts (*Anastasiadou, Jacob & Slack, 2018*). lincRNAs utilize a variety of mechanisms to translationally modulate protein expression, degradation and modification, of which the most critical regulation is competing endogenous RNAs (ceRNAs) theory proposed by *Salmena et al. (2011)*. The hypothesis of ceRNA described complex posttranscriptional communication network of all transcript RNA species including lincRNAs, which can act as natural miRNA sponges to inhibit miRNA functions by sharing miRNA response elements (MRE) (*Giroud & Scheideler, 2017*). Emerging evidence has confirmed the importance of the ceRNA regulatory network of lincRNA-miRNA-mRNA in the initiation and progression of ESCC (*Li et al., 2017*; *Shang et al., 2018*; *Yang et al., 2016*).

Transcriptional factor (TF) is a group of protein molecules that can specifically bind to the certain sequence of the 5′ end of the gene, thereby ensuring that the target gene is expressed in a specific time and space with a specific intensity (*Battaglia, Maguire & Campbell, 2010*). Normal transcription displays a high degree of flexibility over the choice, timing and magnitude of mRNA expression levels, which determines various biological processes including cancer (*Zhong & Ye, 2014*). A number of transcriptional factors have been suggested to participate in the regulation of ESCC (*Pan et al., 2017*; *Zhu et al., 2014*; *Zhu et al., 2016*). But no specific TF regulatory network has been drawn to outline the whole complex processes of nCRT response determination in ESCC.

Although nCRT followed by surgery has shown great advantages, the clinical outcomes of ESCC nCRT are heterogeneous (*Rohatgi et al., 2005*). Favourable survival is consistently achieved in patients who obtain pathologic complete response instead of the nonresponders (*Donahue et al., 2009*). Thus, understanding the underlying determinants of nCRT response could largely improve ESCC management. Despite considerable improvement in the understanding of lincRNAs and miRNAs, only a fraction of annotated ones has been well identified for biological function in ESCC. Until now, the whole picture of the ceRNA regulatory mechanism of lincRNA-miRNA-mRNA in the determination of response of

ESCC to nCRT still remains unclear. In addition, the complete TF regulation network of nCRT response in ESCC is also elusive. In this study, we used RNA microarray data to screen significantly altered lincRNAs, miRNAs and mRNAs in different nCRT responses of ESCC. Besides, the ceRNA modulation and transcriptional factor regulation networks were constructed, which would elucidate the molecular mechanisms involved in the determination of nCRT response of ESCC, thus providing novel clues for ESCC treatment.

## MATERIALS AND METHODS

### Microarray data

Microarray data were obtained from Gene Expression Omnibus (GEO, https://www.ncbi.nlm.nih.gov/geo/) in NCBI (The National Center for Biotechnology Information). GSE59974 is a miRNA expression array of esophageal squamous cell carcinomas with different neoadjuvant chemoradiotherapy response. GSE45670 is an mRNA profiling dataset of esophageal squamous cell carcinomas with different neoadjuvant chemoradiotherapy response. Both GSE59974 and GSE45670 contained 17 responder samples and 11 nonresponder samples. Patients who received neo-CRT and surgery and had fresh pretreatment tissue specimens available for microarray were included. Patients with stage IIB-III squamous cell carcinoma according to the 6th AJCC Cancer Staging, which were technically resectable and medical operable, were considered to receive neo-CRT and surgery. Moreover, patients concurrently received vinorelbine (25 mg/m2) via intravenous injection on days 1, 8, 22, and 29, and cisplatin (75 mg/ m2) via IV on days 1 and 22, or cisplatin (25 mg/m2) via IV daily on days 1 to 4 and 22 to 25. Approximately 4 to 6 weeks after the end of CRT, patients underwent attempted surgical resection of the primary tumor and regional nodes.

### Data processing

GEO2R (http://www.ncbi.nlm.nih.gov/geo/geo2r/) is an interactive tool which can be used to compare two or more groups of samples to identify differentially expressed items in GEO series. We adopted GEO2R to filter differentially expressed genes (DEGs) between responder and non-responder samples separately in each of the data sets. $P < 0.05$ and $|logFC| > 1.5$ were considered as statistically significant. Duplicate gene probes and unspecific probes will be removed. In order to ensure the accuracy of the assay, we selected miRNAs with multiple probes and consistent expression trends as differential miRNAs.

### Gene ontology and pathway enrichment analyses

*Anonymous (2006)* is a major bioinformatics tool for annotating genes and gene products. It contains terms under three categories: cellular component, molecular function, and biological process (2006). Kyoto Encyclopedia of Genes and Genomes (KEGG) is a collection of databases that contain information about genomes, biological pathways, diseases, and chemical substances (*Kanehisa & Goto, 2000*). In the present study, GO and KEGG pathway enrichment analyses were performed by using R package clusterprofiler. $P < 0.05$ was considered as statistically significant.

## Transcription factor regulation network

Search Tool for the Retrieval of Interacting Genes (STRING) database was used to find interacting proteins between different genes (*Von Mering et al., 2005*). Interactions with a combined score >0.4 were defined as significant. Next, we use CentiScape, an application in Cytoscape, to screen for the hub protein. TFCheckpoint database was used to identify the transcription factor among differentially expressed genes. The transcription factor with connection numbers over 5 were considered as the hub transcription factors. Using Cytoscape, we finally built transcription factor regulation network.

## MiRNA regulatory network

The target genes of differentially expressed miRNAs were predicted by miRWalk (http://zmf.umm.uni-heidelberg.de/apps/zmf/mirwalk/index.html), which is a comprehensive online algorithm that provides information on miRNA from Human, Mouse and Rat on their predicted as well as validated binding sites on their target genes (*Dweep et al., 2011*). Accurate classification of 3p/5p for miRNAs was adopted in target gene prediction. The predicted target genes were matched to the genes whose mRNA expressions were opposite to the miRNA profile because miRNAs are negatively regulated genes. In addition, the miRNAs with connection numbers more than 5 were selected as the hub miRNAs.

## CeRNA regulation network

Using miRWalk, we predicted the interaction of lincRNA with miRNA. Then, we matched the targeted lincRNAs with the differentially expressed mRNAs in the mRNA expression array. In this study, the top 10 miRNA was used the built the ceRNA regulation network. According to the ceRNA theory that lincRNAs act as natural miRNA sponges to inhibit miRNA functions, the expressions of lincRNA-miRNA and miRNA-mRNA were all negatively correlated. Finally, the ceRNA regulatory network of lincRNA-miRNA-mRNA in nCRT response of ESCC was constructed by Cytoscape software. The significantly altered miRNAs/mRNAs were analyzed for predictive power in separating responders from non-responders.

# RESULTS

## Identification of DEGs in esophageal squamous cell carcinomas with different neoadjuvant chemoradiotherapy response

The flow chart to visualize the analyses and define/explain the input and output was summarized in Fig. S1. After differential expression analysis, a total of 61 genes in GSE45670 showed decreased expression in responder group and 84 genes were up-regulated in the responder group (Fig. S2). As for the GSE59974 dataset, a total of 247 differentially expressed miRNAs were identified, 110 of which were down-regulated and 137 were up-regulated in responders (Fig. S3). The heatmap of the two arrays were shown in Fig. 1. Heatmap of the hub genes in the regulation network were shown in Fig. S4. All differential expression genes were shown in Table S1.
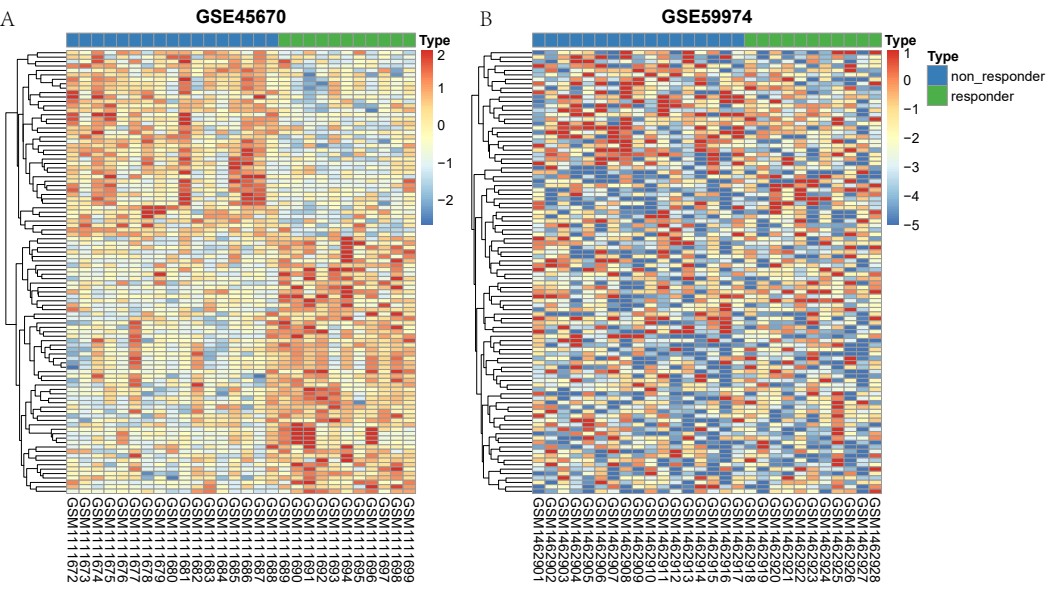

**Figure 1** Heatmap representing the expression levels of genes between responder and non-responder samples. (A) heatmap of the top 100 mRNA of GSE45670. (B) heatmap of the top 100 miRNA in GSE59974.

## GO and pathway functional enrichment analyses

We then performed GO functional enrichment analysis of differentially expressed genes between responders and non-responders. The results showed that the differentially expressed gene mainly enriched in biological process (BP) including positive regulation of cell morphogenesis involved in differentiation, negative regulation of Wnt signaling pathway and positive regulation of cell development. Cellular component (CC) analysis indicated enrichment in proteinaceous extracellular matrix, cornified envelope and basolateral plasma membrane. Besides, these DEGs showed significant enrichment in molecular function (MF) of glycosaminoglycan binding, metalloendopeptidase inhibitor activity and growth factor activity (Table 1, Fig. 2). KEGG analysis enriched these DEGs in pathways of Neurotrophin signaling pathway, Cell adhesion molecules and Wnt signaling pathway (Table 2, Fig. 3).

## Transcription factor regulation network

According to STRING dataset, totally 49 proteins interacted with each other. Using CentiScape software, we selected BMP4 as the hub transcription factors (Table 3). We then built the transcription factor regulation network. As was shown in Fig. 4, the red node represents the up-regulated gene and the blue node represents the down-regulated gene. The darker the color, the higher the expression level was. Purple border represented transcription factors. The node size increased with degree.

## MiRNA regulation network

In the miRNA regulation network, a total of 13 genes were regulated by up-regulated hub miRNAs (Fig. 5A), and 31 genes were regulated by the hub miRNAs with decreased

**Table 1** **Gene ontology analysis of differentially expressed genes associated with response of esophageal squamous cell carcinomas to neoadjuvant chemoradiotherapy.** Results of the top five biological process, cellular component and molecular function.

| Category | GO ID | Term | Count | % | P |
|---|---|---|---|---|---|
| BP | GO:0044236 | multicellular organism metabolic process | 6 | 5.71 | 0.000198 |
| BP | GO:0010770 | positive regulation of cell morphogenesis involved in differentiation | 5 | 4.76 | 0.001407 |
| BP | GO:0030178 | negative regulation of Wnt signaling pathway | 4 | 3.81 | 0.034422 |
| BP | GO:0010720 | positive regulation of cell development | 8 | 7.62 | 0.007031 |
| BP | GO:0042307 | positive regulation of protein import | 3 | 2.86 | 0.02106 |
| CC | GO:0005578 | proteinaceous extracellular matrix | 9 | 8.04 | 0.000389 |
| CC | GO:0001533 | cornified envelope | 4 | 3.57 | 0.000622 |
| CC | GO:0016323 | basolateral plasma membrane | 5 | 4.46 | 0.008398 |
| CC | GO:0030424 | axon | 7 | 6.25 | 0.009971 |
| CC | GO:0055037 | recycling endosome | 4 | 3.57 | 0.015016 |
| MF | GO:0005539 | glycosaminoglycan binding | 7 | 6.80 | 0.000223 |
| MF | GO:0008191 | metalloendopeptidase inhibitor activity | 2 | 1.94 | 0.003633 |
| MF | GO:0033612 | receptor serine | 2 | 1.94 | 0.003633 |
| MF | GO:0008083 | growth factor activity | 4 | 3.88 | 0.017013 |
| MF | GO:0031406 | carboxylic acid binding | 4 | 3.88 | 0.019548 |

**Notes.**
BP, Biological Process; CC, Cellular Component; MF, Molecular Function.

expression (Fig. 5B). Subsequently, we selected the miRNAs with connection numbers >5 as the hub miRNA, which were hsa-miR-520a-3p, hsa-miR-548am-3p, hsa-miR-3184-5p, hsa-miR-548d-5p, hsa-miR-4725-3p, hsa-miR-148a-5p, and hsa-miR-4659a-3p (Table 3).

## CeRNA regulation network

In total, 15 non-coding RNAs showed significant differences in the expression profiling, 7 of which were up-regulated while the others were down-regulated. After matching the predicted results, altogether 12 lincRNAs were associated with the differentially expressed miRNAs (Table 4). APCDD1L-AS1 had the highest degree in the overexpression group while degree of SOX2-OT was the highest in the decreased expression group. The ceRNA regulatory network of lincRNA-miRNA-mRNA in nCRT response of ESCC was shown in Fig. 6. In addition, the AUC (area under curve), sensitivity and specificity of significantly altered miRNA/mRNAs were summarized in Table 5.

## DISCUSSION

Although nCRT before surgery has been accepted as standard strategy for locally advanced ESCC, it is still difficult for surgeons to predict whether the patient present favourable response to nCRT. In order to understand the underlying determinants of nCRT response of ESCC patients, miRNA expression array (GSE59974) and mRNA profiling dataset (GSE45670) of ESCC with different nCRT responses were systematically analyzed. The pretreatment samples were adopted for microarray, the results of which indicate predictive network for nCRT response. Finally, the ceRNA modulation and transcriptional factor

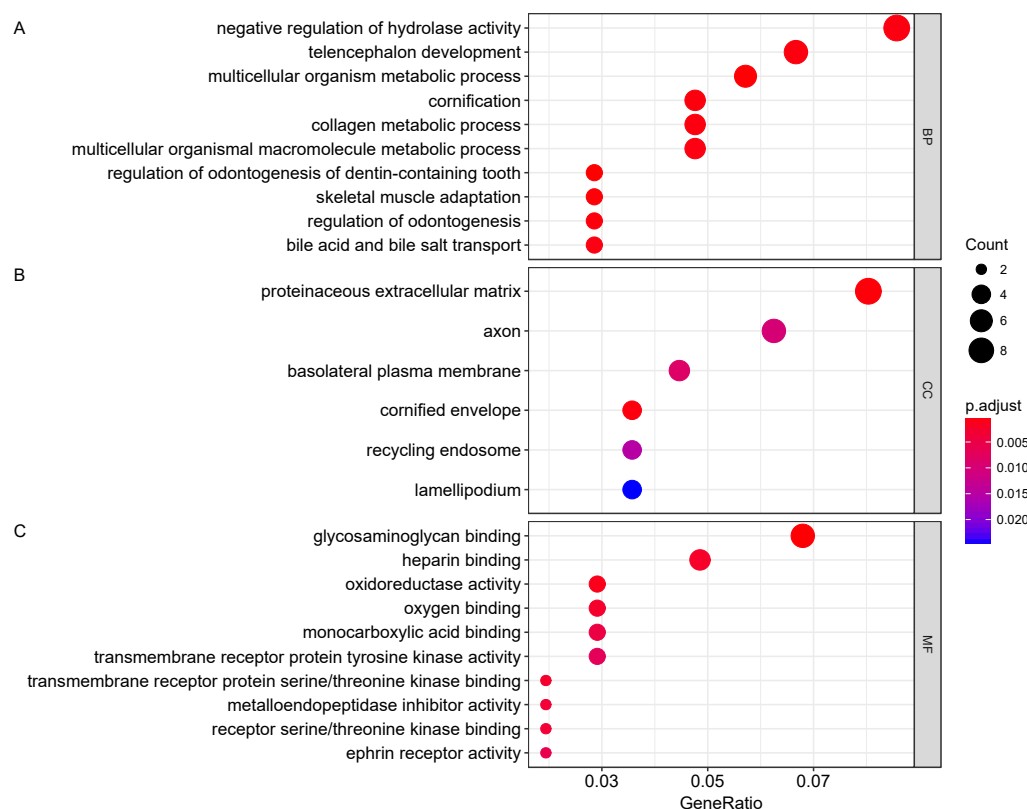

**Figure 2** **Gene ontology analysis of differentially expressed genes associated with response of esophageal squamous cell carcinomas to neoadjuvant chemoradiotherapy.** A total of 145 mRNAs were chosen for GO analysis for biological process, cellular component and molecular function. The size of circle represented the gene counts in go terms, the colour represented the *p*-value of go terms. (A) biological process. (B) cellular component. (C) molecular function.

**Table 2** **KEGG analysis for differentially expressed genes associated with response of esophageal squamous cell carcinomas to neoadjuvant chemoradiotherapy.** Result of the pathway analysis of differentially expressed genes.

| ID | Description | Count | % | P | Genes |
|---|---|---|---|---|---|
| hsa04360 | Axon guidance | 5 | 12.19512 | 0.002621 | EPHB2, L1CAM, CAMK2B, NRP1, PAK5 |
| hsa00980 | Metabolism of xenobiotics by cytochrome P450 | 3 | 7.317073 | 0.007782 | AKR1C1, EPHX1, ADH1B |
| hsa04722 | Neurotrophin signaling pathway | 3 | 7.317073 | 0.027708 | NGF, CAMK2B, RPS6KA6 |
| hsa04514 | Cell adhesion molecules (CAMs) | 3 | 7.317073 | 0.045651 | CLDN8, L1CAM, CDH2 |
| hsa04310 | Wnt signaling pathway | 3 | 7.317073 | 0.046434 | CAMK2B, WIF1, SOST |

regulation networks were constructed, which would contribute to the understanding of molecular mechanisms involved in the determination of nCRT response of ESCC.

A total of 61 genes showed decreased expression in responder group and 84 genes were up-regulated in the responder group in the present study. According to the functional enrichment analysis, DEGs between responders and non-responders were mainly enriched in biological process including regulation of cell morphogenesis involved in differentiation, Wnt signaling pathway, cell development and protein import. Additionally, these DEGs

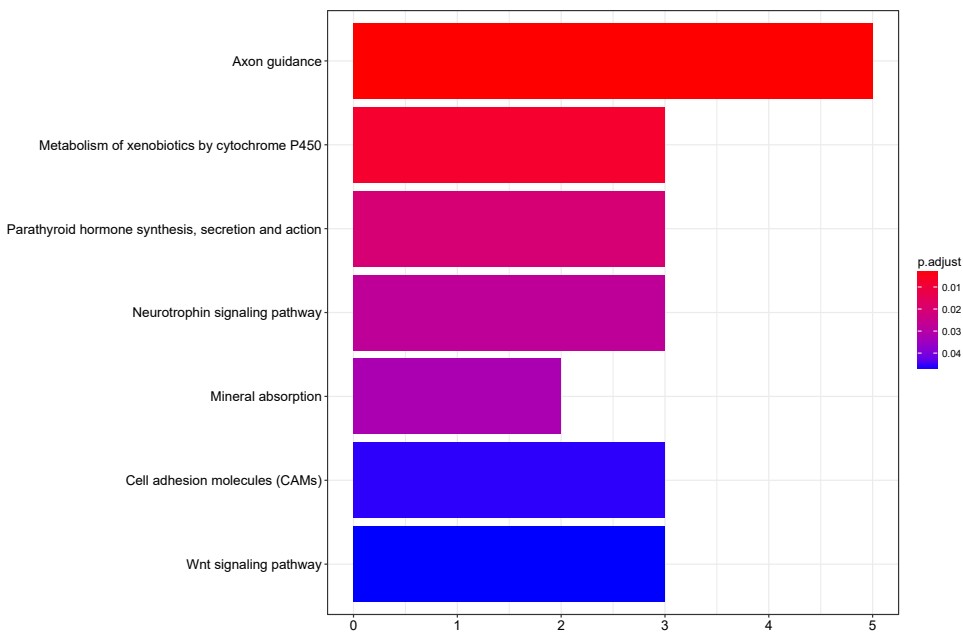

**Figure 3** **KEGG pathway analysis of differentially expressed genes associated with response of esophageal squamous cell carcinomas to neoadjuvant chemoradiotherapy.** A total of 145 mRNAs were chosen for kegg analysis. the color represented the *p* value of kegg terms.

**Table 3** **Hub transcription factor and miRNA associated with response of esophageal squamous cell carcinomas to neoadjuvant chemoradiotherapy.** One gene and seven miRNAs were selected as hubs.

| gene | ID | Targeting mRNAs | Change |
|---|---|---|---|
| BMP4 | 652 | ZIC1, SOST, ALB, TF, CDH2, PTHLH | Down |
| hsa-miR-520a-3p | MIMAT0002834 | BAGE, ELAVL2, GPR158, INHBA, KLK5, PTHLH | Down |
| hsa-miR-548am-3p | MIMAT0019076 | CTAG2, DNAH5, GLCCI1, QSOX2, RHEBL1, RSPO2, SEZ6L2 | Down |
| hsa-miR-3184-5p | MIMAT0015064 | C1QTNF6, CDSN, IQGAP2, RRBP1, SEZ6L2, TNNT1, WFDC12 | Down |
| hsa-miR-548d-5p | MIMAT0004812 | CDH18, COL12A1, DDAH1, KRT37, PLEKHS1, PPP4R4, PSG5, SKP2 | Down |
| hsa-miR-4725-3p | MIMAT0019844 | C1QTNF6, EPHA10, FCHSD1, PLEC, PLEKHS1, QSOX2, RHEBL1, WFDC12 | Down |
| hsa-miR-148a-5p | MIMAT0004549 | ABCA13, ADH1B, KLF12, NKAIN2, SPOCK3, WIF1 | Up |
| hsa-miR-4659a-3p | MIMAT0019727 | CHP2, CLDN8, CYP26A1, FAXC, JAKMIP3, KLF12, MBNL1, SCUBE2 | Up |

showed significant enrichment in molecular function of glycosaminoglycan binding, metalloendopeptidase inhibitor activity, receptor serine, growth factor activity and carboxylic acid binding. KEGG analysis enriched these DEGs in pathways of axon guidance, metabolism of xenobiotics by cytochrome P450, neurotrophin signaling pathway, cell adhesion molecules and Wnt signaling pathway. Wnt pathway has previously been reported to play critical role in cell fate specification, cell proliferation and cell migration during

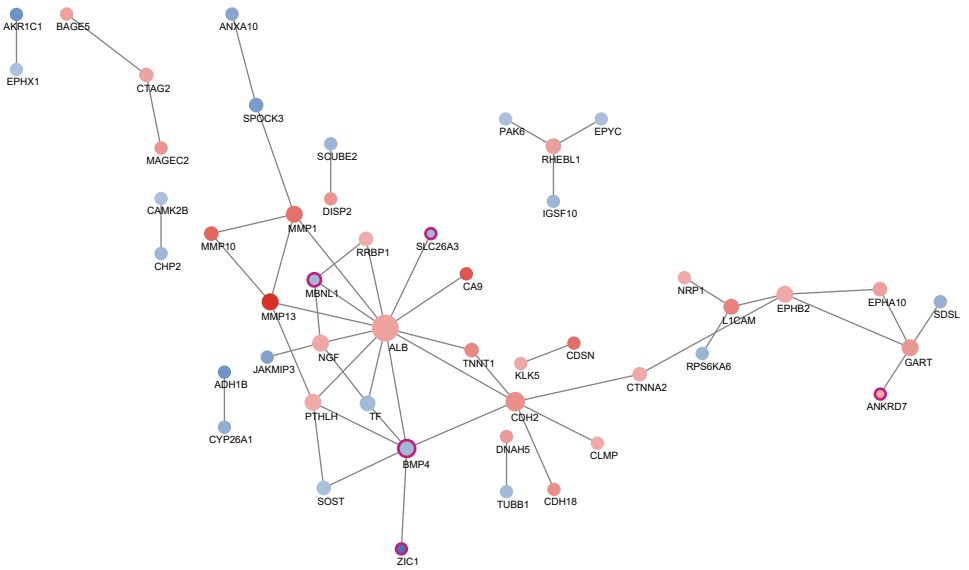

**Figure 4** **Protein-protein interaction between differentially expressed genes.** A total of 49 proteins interacted with each other. The size of the circle in the network represents the number of connections, red represents an up-regulated gene, blue represents a down-regulated expression gene, and purple frame represents a transcription factor.

A                                                           B

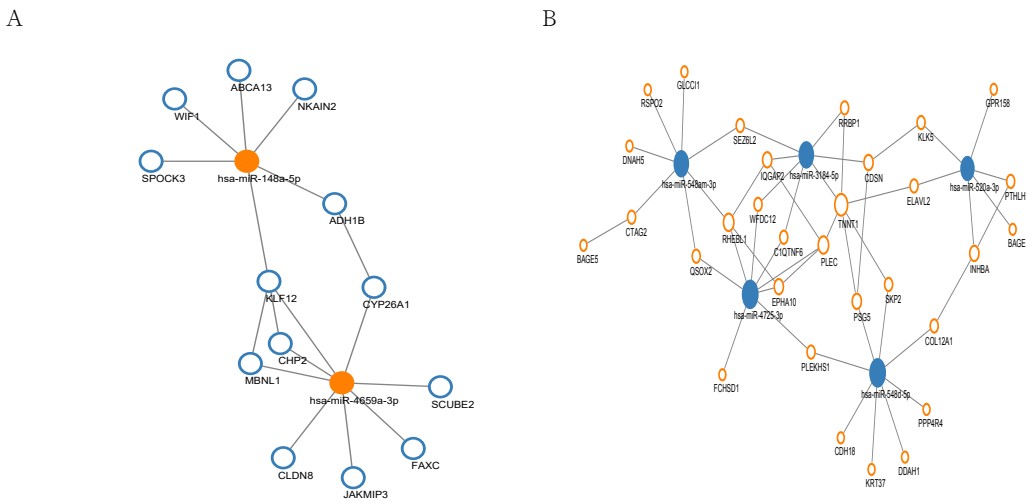

**Figure 5** **MiRNA regulation network associated with response of esophageal squamous cell carcinomas to neoadjuvant chemoradiotherapy.** The size of the circle in the network represents the number of connections. pink represents high expression, blue represents low expression. The fill color represents mRNA. Outer frame color represents miRNA. (A) 13 genes were regulated by two highly expressed hub miRNAs. (B) 31 genes were regulated by the 5 low expressed hub miRNAs.

embryonic development and carcinogenesis (*Duchartre, Kim & Kahn, 2016*). Growth factor is a naturally occurring substance capable of stimulating cellular growth, proliferation, healing, and cellular differentiation (*Lappano et al., 2013*). The results of this study reveal

**Table 4  Differentially expressed lncRNA associated with response of esophageal squamous cell carcinomas to neoadjuvant chemoradiotherapy.** Nine lincRNAs were selected as hub lncRNAs.

| Gene name | ID | Chromosome | Gene type | Targeting miRNAs | Change | Degree |
|---|---|---|---|---|---|---|
| APCDD1L-AS1 | ENSG00000231290 | 20 | processed_transcript | hsa-miR-1224-3p, hsa-miR-520a-3p, hsa-miR-19b-1-5p, hsa-miR-4708-5p, hsa-miR-4724-5p, hsa-miR-4725-3p | Up | 29 |
| ERVMER61-1 | ENSG00000230426 | 1 | lincRNA | hsa-miR-374a-3p, hsa-miR-19b-1-5p, hsa-miR-4724-5p, hsa-miR-548am-3p, hsa-miR-548d-5p | Up | 16 |
| LINC00598 | ENSG00000215483 | 13 | lincRNA | hsa-miR-520a-3p, hsa-miR-4708-5p, hsa-miR-4725-3p, hsa-miR-548d-5p | Up | 18 |
| LINC01553 | ENSG00000235931 | 10 | lincRNA | hsa-miR-2115-3p, hsa-miR-374a-3p, hsa-miR-520a-3p, hsa-miR-19b-1-5p, hsa-miR-4708-5p, hsa-miR-548am-3p, hsa-miR-548d-5p | Up | 26 |
| MCHR2-AS1 | ENSG00000229315 | 6 | antisense | hsa-miR-1224-3p, hsa-miR-19b-1-5p, hsa-miR-4708-5p, hsa-miR-4724-5p | Up | 12 |
| ZFHX4-AS1 | ENSG00000253661 | 8 | antisense | hsa-miR-1224-3p, hsa-miR-2115-3p, hsa-miR-548d-5p | Up | 14 |
| ADAMTS9-AS2 | ENSG00000241684 | 3 | antisense | hsa-miR-4760-5p, hsa-miR-148a-5p, hsa-miR-204-3p, hsa-miR-3617-5p, hsa-miR-561-3p, hsa-miR-3689a-5p, hsa-miR-4659a-3p, hsa-miR-548at-5p, hsa-miR-593-3p | Down | 48 |
| KRTAP5-AS1 | ENSG00000281801 | 11 | antisense | hsa-miR-3617-5p, hsa-miR-4446-3p, hsa-miR-4659a-3p | Down | 22 |
| LINC00664 | ENSG00000268658 | 19 | lincRNA | hsa-miR-148a-5p, hsa-miR-204-3p, hsa-miR-3617-5p, hsa-miR-3689a-5p, hsa-miR-4446-3p, hsa-miR-4659a-3p, hsa-miR-548at-5p | Down | 36 |
| LINC00942 | ENSG00000249628 | 12 | lincRNA | hsa-miR-204-3p, hsa-miR-3617-5p | Down | 22 |
| LINC00960 | ENSG00000242516 | 3 | lincRNA | hsa-miR-148a-5p, hsa-miR-3617-5p, hsa-miR-561-3p, hsa-miR-4446-3p, hsa-miR-4659a-3p, hsa-miR-548at-5p, hsa-miR-593-3p | Down | 27 |
| SOX2-OT | ENSG00000242808 | 3 | sense_overlapping | hsa-miR-148a-5p, hsa-miR-204-3p, hsa-miR-3617-5p, hsa-miR-561-3p, hsa-miR-3689a-5p, hsa-miR-4659a-3p, hsa-miR-548at-5p, hsa-miR-593-3p | Down | 56 |

promising novel functions of these substance or pathways in determining nCRT response of ESCC.

Transcriptional factor (TF) is a group of protein molecules that can efficiently control the expression of target gene in a specific time and space with a specific intensity (*Swift & Coruzzi, 2017*). With the help of STRING and CentiScape software, we built the transcription factor regulation network which determines response of ESCC to neoadjuvant chemoradiotherapy. As was shown in the TF regulation network, TFs such as MBNL1, SLC26A3, BMP4, ZIC1 and ANKRD7 synergistically modulate the response of ESCC patients to nCRT. It was worth noting that BMP4 was the hub transcription factor in this
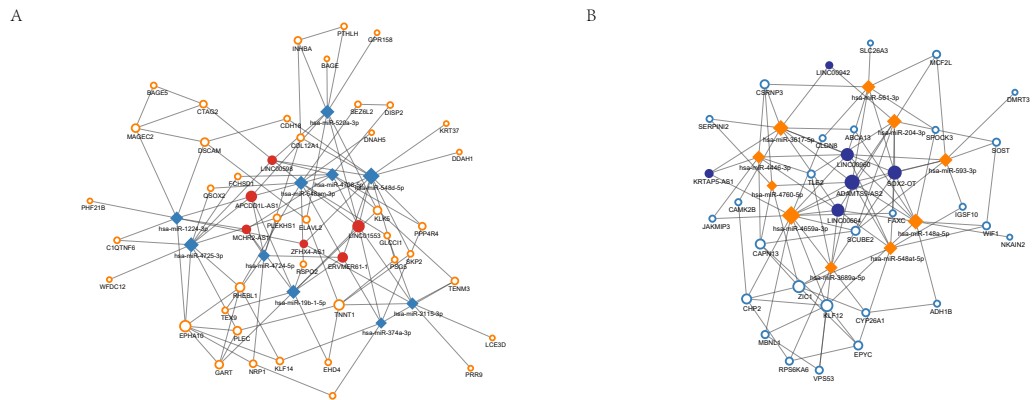

**Figure 6** **Competitive endogenous RNA (ceRNA) regulation network associated with response of esophageal squamous cell carcinomas to neoadjuvant chemoradiotherapy.** The size of the circle in the network represents the number of connections. pink represents high expression, blue represents low expression. Diamonds represent miRNAs. The fill color of the circle represents lncRNA. The outer frame color represents mRNA. (A) 6 highly expressed lncRNAs regulate 10 miRNAs and 44 genes. (B) 6 highly expressed lncRNAs regulate 10 miRNAs and 27 genes.

**Table 5** **The top 5 predictive power of hub miRNA/mRNAs associated with response of esophageal squamous cell carcinomas to neoadjuvant chemoradiotherapy.** The area under curve, sensitivity and specificity of significantly altered miRNA/mRNAs were summarized.

| Symbol | Auc | Specificity | Sensitivity |
|---|---|---|---|
| hsa-miR-4460 | 0.909 | 0.765 | 1 |
| hsa-miR-369-3p | 0.906 | 0.765 | 0.909 |
| hsa-miR-581 | 0.898 | 1 | 0.818 |
| hsa-miR-4524a-5p | 0.898 | 0.941 | 0.818 |
| hsa-miR-548ap-3p | 0.893 | 0.765 | 1 |
| QSOX2 | 0.92 | 1 | 0.727 |
| AZIN2 | 0.914 | 0.765 | 0.909 |
| RELL2 | 0.914 | 0.941 | 0.909 |
| EPHB2 | 0.914 | 0.765 | 0.909 |
| MEAF6 | 0.914 | 0.882 | 0.818 |

regulation network with the highest degree. Previously, BMP4 was found to be significantly more up-regulated in ESCC than normal squamous epithelium (*Van Baal et al., 2008*). In addition, BMP signaling pathways has been reported to affect migration and invasion of esophageal squamous cancer cells differently (*Hu et al., 2017*). In this study, our findings indicated that BMP4 might also play an important role in ESCC nCRT response. Most other TFs have never been linked to ESCC, which offers us potential therapeutic targets to improve the nCRT response of ESCC.

By means of miRWalk database, we constructed miRNA regulation network involved in the nCRT response of ESCC including 13 genes regulated by up-regulated hub miRNAs and 31 genes regulated by the hub miRNAs with decreased expression. The miRNAs with connection numbers over 5 were defined as hub miRNAs, which were hsa-miR-520a-3p,

hsa-miR-548am-3p, hsa-miR-3184-5p, hsa-miR-548d-5p, hsa-miR-4725-3p, hsa-miR-148a-5p, and hsa-miR-4659a-3p. Previously, miR-520a has been proved to modulate the expression of ErbB4 and suppresses the proliferation and invasion of ESCC cells in vitro, indicating its role as a tumor suppressor (*Ye et al., 2014*). Besides, miR-148a was indicated to be implicated in carcinogenesis in primary ESCC through regulating HLA-G expression (*Chen, Luo & Zhang, 2017*). Together with our findings in this study, miR-520a and miR-148a might serve as key regulators of ESCC carcinogenesis and nCRT response.

In order to construct the ceRNA regulatory network of lincRNA-miRNA-mRNA, we next predicted the interaction of miRNAs with lincRNAs and mRNAs. According to the ceRNAs theory, the expressions of miRNAs should be negatively correlated with expressions of targeting lincRNAs and mRNAs. Therefore, we overlapped the predicted targets of up-regulated Differentially Expressed miRNAs (DEMs) with down-regulated Differentially Expressed lincRNAs (DELs) and DEGs, as well as overlapped the predicted targets of down-regulated DEMs with up-regulated DELs and DEGs. Finally, the ceRNA regulatory network of lincRNA-miRNA-mRNA in determining nCRT response of ESCC was built, which contains 12 lincRNAs associated with the DEMs. APCDD1L-AS1 had the highest degree in the overexpression group while degree of SOX2-OT was the highest in the decreased expression group. These lincRNAs hold great potential as biomarkers of predicting nCRT response of different ESCC patients.

During the past few years there has been considerable progress in ESCC, where it has been most extensively studied in nCRT therapy followed by surgery. However, limited information was revealed about determination of nCRT response and few validated biomarkers could guide ESCC chemotherapy or radiotherapy presently. The identified significantly altered lincRNAs, miRNAs and mRNAs involved in the nCRT response of ESCC in this study would offer physicians opportunities to save ineffective treatments and turn to alternative methods, thus avoiding the over- or under-treatment of ESCC individuals. In addition, the ceRNA regulatory network of lincRNA-miRNA-mRNA and transcriptional factor regulatory network we constructed would shed new light on the molecular mechanisms determining nCRT response of ESCC. We should acknowledge the limitation that no experiment was performed to confirm the findings, which should be researched in the future.

## CONCLUSION

We identified a series of significantly altered lincRNAs, miRNAs and mRNAs involved in the nCRT response of ESCC by comparing RNA microarray data of responders and non-responders. In addition, the ceRNA regulatory network of lincRNA-miRNA-mRNA and transcriptional factor regulatory network were constructed, which would elucidate novel molecular mechanisms determining nCRT response of ESCC, thus providing promising clues for clinical therapy. Core regulatory miRNAs of miR-520a, miR-548am, miR-3184, miR-548d, miR-4725, miR-148a, miR-4659a and key regulatory transcriptional factors of MBNL1, SLC26A3, BMP4, ZIC1 and ANKRD7 might provide biological insight into the full repertoire of nCRT response of ESCC.

### Funding

This study was supported by Natural Science Foundation of Liaoning Province (grant no. 2015020561) and the Fund for Scientific Research of the First Hospital of China Medical University (grant no. fsfh1514). The funders had no role in study design, data collection and analysis, decision to publish, or preparation of the manuscript.

### Grant Disclosures

The following grant information was disclosed by the authors:
Natural Science Foundation of Liaoning Province: 2015020561.
Fund for Scientific Research of the First Hospital of China Medical University: fsfh1514.

### Competing Interests

The authors declare there are no competing interests.

### Author Contributions

- Mingrui Shao performed the experiments, analyzed the data, contributed reagents/materials/analysis tools, prepared figures and/or tables.
- Wenya Li conceived and designed the experiments, authored or reviewed drafts of the paper, approved the final draft.

### Data Availability

The microarray data are available at GEO: GSE59974 and GSE45670.

The raw measurements are available in Table S1. The raw data shows all the differentially expressed genes in this study.

### Supplemental Information

Supplemental information for this article can be found online at http://dx.doi.org/10.7717/peerj.6668#supplemental-information.

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
