# Peer review of "Transcriptional factor regulation network and competitive endogenous RNA (ceRNA) network determining response of esophageal squamous cell carcinomas to neoadjuvant chemoradiotherapy"

_PeerJ, doi:10.7717/peerj.6668_

## Round 0.1 · original submission · Major Revisions

I have received three independent expert reviews. All reviewers have indicated that they can see the usefulness of your research. Two of the three reviewers have made a number of specific and useful comments that I expect you to accept and respond to appropriately and positively in your revised manuscript.

From an editorial standpoint, please correct the language and use of abbreviations as indicated by the reviewer examples (although these are not comprehensive and thorough checking is required throughout the manuscript).

Reviewer 1 ·

Basic reporting

1. The English should be improved, for example, accurately use the abbreviations.
2. More details of the status of nCRT in ESCC should be introduced.
3. The figure legend should be detailed. Presently it is too simple to describe the figures. Figure 1 is confusing, which samples are responder samples. The number of samples in figure1A is not consistent with the manuscript. No figure 6 is shown.

Experimental design

1. More information of the two microarray data should be described.

Validity of the findings

no comment

Additional comments

This article shows the TF regulation network and ceRNA network determining response of ESCC to nCRT. nCRT followed by surgery benefits survival for patients with ESCC, but the clinical outcomes of nCRT are heterogeneous. The theme is interesting and this analytical work can increase understanding of molecular mechanisms determining nCRT response of ESCC.

·

Basic reporting

The manuscript is well structured and the written english is clear. The lack of molecular biomarkers of neoadjuvant chemoradiotherapy response in esophageal cancer is a relevant theme in the oncology field and the authors provide a good background to explain the need of more knowledge in this area. The number of literature references also seem suitable for this type of research article.

Experimental design

The research question is well defined, and fits the aims and scope of the journal.
In terms of study design and methodology, the study was well panned and the methods were described with sufficient detail.
The authors should clarify the "Materials and Methods" section in the Abstract since they say that "RNA sequencing data of GSE59974 and GSE45670 were analyzed..." and then in the "Material and Methods" section of the manuscript they talk about "Microarray data". Since sequencing data is different from microarray data the authors should correct this.

Validity of the findings

The results were very interesting and they should be validated in esophageal cancer patients in the future.

Additional comments

The manuscript was very good and covered an emergent theme in the oncology field, I recommend its publication.

Reviewer 3 ·

Basic reporting

Overall, the manuscript is a well designed observational study but needs some work in the following areas
1) There are several times when the authors use the terms genes and mRNA/lincRNA or miRNA interchangeably. Please use mRNA or miRNA or lincRNA consistently
2) Several sentences are incomplete fragments that can limit reader understanding. Please revise (e.g line 210-212 "And growth factor....")
3) The figure legends are unsatisfactory. E.g. Figure 1 just says heatmap of two arrays without a color key, or sample description. Additionally, the reader needs to go back to materials and methods to figure out the array numbers (Viz. GSE 45670 and GSE 59974) . Please include descriptions that are more comprehensive
4) Given that neoadjuvant chemoradiotherapy(nCRT) is the subject of the review, very little background or details have been provided on this topic.
5) Tables captions need work. E.g. Table 1 uses acronyms that are not defined in legend (BP,CC MF)

Experimental design

This manuscript is within the scope of the journal and the methods are described with clarity
While the research question is defined, the authors have not contextualized their findings in the results or discussion. They need to work on conveying the significance of the results( see more in note to authors)

Validity of the findings

The findings while valid provide no clear biomarkers to separate responders from non-responders to nsCRT nor do the authors provide any further analyses to support their applicability. Some tools to enhance the value of the findings include
1) calculating the predictive power of the suggested biomarkers in separating responders from non-responders,
2) Making a heatmap of mRNA, miRNA and lincRNA identified in your network to compare responders and non-responders
3) Discuss the results from theses analyses and any additional data that might supplement this work
This will help answer their research question of using RNA transcripts to determine response to ESCC more definitively.

Additional comments

I believe that the authors have made a strong attempt to understand the interactions between mRNA-miRNA and lnRNA expression in Esophageal squamous cell carcnomas(ESCC).
Overall, I think that the readability and applicability of the paper will be greatly improved by focusing
1) The questions that the authors are trying to address: are you identifying prognostic markers or performing a RNA network analysis?
2) The results section needs to be explained with more clarity. There are several instances where the authors say 'x' genes were identified without any explanation as to how that number was reached. Furthermore, the authors need to explain how they are selecting genes from one analyses such as DEG and using them for the next step like network building. One suggestion would be to create a flow chart to visualize the analyses and define/explain the input and output of each tool with it.
3) The authors dont explain the significance of any of their findings. e.g What does it mean to have 'x' genes be significant by this tool? Why is this particular analyses necessary. The Gene ontology analyses are just stated without any explanation of the data
Some additional points include
- Please provide a citation for STRING and the significance cut off for interactions in the materials and methods (lines 121-123).
- What was the input for STRING that resulted in the identification of 49 interacting proteins (Line 168)
- Could the authors use GSE numbers in conjunction with type of array so the reader can follow the results and discussion without needing to figure out the array type being referred to?
- Both the supplementary figures lack a legend, figure titles and a corresponding explanation in the text.
- In figure 1; the heatmap needs a color key, and a horizontal bar over it showing which samples are responders and which are non-responders. Is the clustering supervised?
-Please provide the differential genes identified in your analyses as supplemental information
- The authors use the terms "highly expressed" and "low expressed" several times (e.g line 171) do you mean up and down regulated? If not define these terms
- The figures say figure 5 twice, replace second instance as fig 6

---

## Round 0.2 · Minor Revisions

Please see the comments from the reviewer and revise your manuscript to explicitly state whether you analysed samples prior to treatment or after they began treatment, and revise the manuscript discussion as suggested.

Please modify Figure 5 to make it easier for the reader to see.

You may wish to include the third comment in a revised discussion.

Reviewer 3 ·

Basic reporting

no comment

Experimental design

no comments

Validity of the findings

No comments

Additional comments

I believe that the authors have sufficiently addressed a majority of my concerns. I request the authors to clarify the following in the manuscript
1) time points at which samples were analyzed. Are your microarray samples from patients at baseline( i.e prior to receiving any nCRT?). If yes, then you are studying predictive networks while if they are post treatment, you are really looking at mechanisms for non-response. Please clarify that in your paper, particularly in the discussion section
2) Figure 5B in particular is hard to read. Could you please upload a higher resolution version of it?
3) In STRING, you have identified a number of “hub” genes including BMP4( discussed), ALB and EPHB2. However, you focus on only BMP2. Is it because it is a transcription factor? If not, could you clarify if there is anything known about the other hubs in ESCC?

---

## Round 0.3 · accepted · Accept

Many thanks for answering the comments and submitting your revised manuscript

#